# A Novel Consensus Fuzzy K-Modes Clustering Using Coupling DNA-Chain-Hypergraph P System for Categorical Data

**Zhenni Jiang and Xiyu Liu \***

Business School, Academy of Management Science, Shandong Normal University, Jinan 250014, China; jennysdnu@163.com
**\*** Correspondence: xyliu@sdnu.edu.cn

**Abstract:** In this paper, a data clustering method named consensus fuzzy k-modes clustering is proposed to improve the performance of the clustering for the categorical data. At the same time, the coupling DNA-chain-hypergraph P system is constructed to realize the process of the clustering. This P system can prevent the clustering algorithm falling into the local optimum and realize the clustering process in implicit parallelism. The consensus fuzzy k-modes algorithm can combine the advantages of the fuzzy k-modes algorithm, weight fuzzy k-modes algorithm and genetic fuzzy k-modes algorithm. The fuzzy k-modes algorithm can realize the soft partition which is closer to reality, but treats all the variables equally. The weight fuzzy k-modes algorithm introduced the weight vector which strengthens the basic k-modes clustering by associating higher weights with features useful in analysis. These two methods are only improvements the k-modes algorithm itself. So, the genetic k-modes algorithm is proposed which used the genetic operations in the clustering process. In this paper, we examine these three kinds of k-modes algorithms and further introduce DNA genetic optimization operations in the final consensus process. Finally, we conduct experiments on the seven UCI datasets and compare the clustering results with another four categorical clustering algorithms. The experiment results and statistical test results show that our method can get better clustering results than the compared clustering algorithms, respectively.

**Keywords:** consensus clustering; fuzzy k-modes algorithm; chain P system; hypergraph structure

## 1. Introduction

Data clustering has recently attracted more attentions in practical applications. However, most studies are about numerical data. In this method, the distance between the clustering center and the data objects are calculated by the standard distance metrics. However, there are many classification datasets that do not have a natural order or distance between the parts. For example, in the real world, each classification attribute of blood type has a unique classification value, such as $[\mathrm{A}, \mathrm{B}, \mathrm{O} \ or \ \mathrm{AB}]$. Therefore, research into categorical data is a difficult and challenging task, which attracts many data mining researchers.

In 1998, Huang [1] proposed the k-modes algorithm for the categorical data clustering. The k-modes algorithm calculated the distance between the object and the cluster center by the Hamming distance instead of Euclidean distance which is used in the k-means algorithm. Then, Huang [2] proposed the fuzzy k-modes algorithm (FKM). This algorithm was an extended version of the k-modes algorithm. Thereafter, many algorithms have been proposed for the clustering of categorical data [3]. These methods were mostly based on numerical data clustering algorithms, such as ROCK [4], CACTUS [5], COOLCAT [6], LIMBO [7], wk-modes [8], MOGA [9], NSGA-FMC [10], SBC [11],

MOFC [12], and so on. The research content is mainly divided into two categories. One is the method of the similarity measure about the categorical data. Hamming distance is the main method for calculating the distance between the data point and the clustering center which measures the distance two different categorical values at 1, otherwise is 0. This method has been used in many other clustering algorithms, such as the k-prototypes algorithm [1] and Squeezer [13]. In addition to the Hamming distance, there are many other distance metrics [14], such as Ahmad's distance metric, the Jaccard metric, the association-based distance metric, and the context-based distance metric. Another direction is utilizing the optimization algorithm to improve the clustering performance. The WFKM algorithm [2] strengthened the basic k-modes clustering by associating higher weights with features useful in analysis. The GFKM algorithm [15] integrated the genetic algorithm into the fuzzy k-modes algorithm, aiming to find the global optimal solution. The IWFKM algorithm [16] replaced the Hamming distance with the frequency probability-based distance metric, which has been proved to improve the clustering results. Apart from this method, the multi-objective clustering algorithm was also considered to improve the performance of the categorical algorithm [17]. Rough set theory was also introduced into the K-modes algorithm, which is used to calculate the density of each candidate modes to characterize the distribution around it [18]. Some researchers developed cluster weighted estimators of marginal proportion, which remain unchanged under the amount of information, and derive the similarity, one sample ratio, goodness of fit and independent chi square test for clustering data [19]. Some research also introduced other evolutionary algorithms into K-modes algorithm. For example, the firefly algorithm was used to generate initial clusters [20]. Some scholars started from the data themselves, and studied the possibility of reducing the dimension of representation based on spatial structure while maintaining the same representation capability [21].

Păun [22] proposed membrane computing (also called P system) in 1998. This aims to abstract computational models from the structure and function of biological cells and the cooperation between organs and tissues. Up to now, membrane computing mainly includes three basic models: cell-like p system, tissue-like p system and neuron-like p system. In the process of calculation, each cell is regarded as an independent unit, and each cell operates independently and does not interfere with each other. The whole membrane system operates in a highly parallel mode. According to the research content about the P system, it can be divided into theoretical study and application study. For the theoretical research, researchers use the direct membrane to solve problems [23]. In this aspect, some new P system models are proposed which can improve the computation power with the min cells or spikes [24,25]. Many other variants of membrane systems were also proposed in [26–29]. In application research, the direct membrane algorithm was used to solve some practical problems by the researchers [30]. Some researchers also used the coupled membrane system to realize the clustering process [31,32]. These membrane systems are all improvements based on cell-like P systems, tissue-like P systems and neuron-like systems. For these kinds of system, they are all designed based on simple structures and can not solve the problems with complicated structure. For example, the traditional P system cannot store multivariate data with complex relationships. Therefore, our team previously proposed the concept of P system with the simple complex structure [33] and chain structure. For instance, Liu and Xue established the new P system based on the simple complex structure in [33], and Luan and Liu designed the chain P system [34], while Yan and Xue proposed the chain-hypergraph P system [35].

Based on the above analysis, we proposed a novel hybrid DNA-chain-hypergraph P system to implement the consensus clustering (DCHP-FCC). The DCHP-FCC system contains three reaction chain membrane subsystems and one consensus subsystem. Three different base clustering algorithms are used in the three subsystems, respectively. This operation combines the different advantages of the three algorithms, and the DNA genetic algorithm implements the consensus clustering process in the consensus system. The experiment on seven UCI datasets is conducted. The experimental results show that our proposed method outperforms the results of the state-of-the-art methods.

This work makes the following contributions:

(1) A novel DCHP system is designed which combines the advantage of the chain structure and hypergraph topology structure. Three reaction chain membrane subsystems and one consensus subsystem are designed to generate the basic partitions and integrate basic partitions, respectively.

(2) A revised k-means which is optimized by the DNA genetic algorithm is used for a basic partition integration strategy which can optimize the initial clustering center and obtain the global optimal solutions.

(3) Simulation is performed using well-known datasets in the UCI machine learning repository to verify the clustering quality of the DCHP-FCC.

The rest of this paper is organized as follows. Section 2 introduces the basic concepts of the k-modes algorithm, consensus clustering and chain and hypergraph structure. The coupling DCHP system is illustrated in Section 3. Experiments and results are analyzed in Section 4. Section 5 summarizes conclusions and future research directions.

## 2. Basic Concepts

### 2.1. Three Basic Fuzzy K-Modes Clustering Algorithms

The fuzzy k-modes algorithm (FKM) was proposed by Huang and Ng [1], and is one of the most popular clustering algorithms for categorical data. This type of method has improved the k-modes algorithm by the corresponding membership degree value in different clustering.

**Definition 1.** *Defining 4-tuple* $S = (U, A, V, f)$ *is an information system, where* $U$ *represents non-empty finite set of objects,* $A$ *refers to the non-empty finite set of attributes,* $V = \cup_{a \in A} V_a$, $V_a$ *records the domain of attribute* $a$, *and* $f : U \times A \to V$ *is a total function such that* $f(u, a) \in V_a$, *for every* $(u, a) \in U \times A$.

Let $X$ be the dataset which has $n$ categorical objects. Each object $x_i$ $(1 \le i \le n)$ has $p$ attributes, so that $x_i = \{x_{i1}, x_{i2}, ..., x_{ip}\}$. The objective of the FKM is characterized as follows:

$$F_{FKM}(U, Z, X) = \sum_{j=1}^{k} \sum_{i=1}^{n} u_{ji}^{\alpha} d(x_i, z_j) \tag{1}$$

subject to:

$$0 \le u_{ji} \le 1, \quad 1 \le i \le n, 1 \le j \le k, \tag{2}$$

$$\sum_{j=1}^{k} u_{ji} = 1, \quad 1 \le i \le n \tag{3}$$

$$0 < \sum_{i=1}^{n} u_{ji} < n, \quad 1 \le j \le k \tag{4}$$

where, $\alpha$ is weight component, $U = (u_{ji})$ is a $k \times n$ matrix which records the fuzzy membership degree, $Z = \{z_1, ..., z_k\}$ is the set of the clustering centers. $X = (X_1, X_2, ..., X_n)$ is the data matrix, where $X_i$ is the $i$th point. $d(x_i, z_j)$ calculates the distance between the object $x_i$ and the clustering center $z_j$. The distance is calculated by simple matching dissimilarity measures or Hamming distance which is showed as follows:

$$d(x_i, z_j) = \sum_{l=1}^{m} \delta(x_{il}, z_{jl}) \tag{5}$$

and

$$\delta(x_{il}, z_{jl}) = \begin{cases} 0, & if \ x_{il} = z_{jl} \\ 1, & if \ x_{il} \neq z_{jl} \end{cases} \tag{6}$$

Based on the proposed scheme, a weight vector is added in the conventional fuzzy k-modes algorithm [2]: $W = [w_1, w_2, ..., w_p]$, where $w_l$ represents the weight for the *l*th variable, $\forall l = 1, 2, ..., p$. So, the objective function of the WFK-modes (WFKM) algorithm is:

$$F^{\alpha, \beta}(U, Z, X, W) = \sum_{j=1}^{n} \sum_{i=1}^{n} u_{ji}^{\alpha} d_{ji}^{W} \tag{7}$$

where, $\beta$ is the power of the attribute weight $w_l$ which is a measure of emphasis on weights. Similar to the FKM algorithm, $d_{ji}^{W} = d^{W}(Z_j, X_i) = \sum_{l=1}^{p} \delta^{W}(z_{jl}, x_{il})$, where $z_{ji}$ is the *l*th term of $Z_j$ and $x_{il}$ refer to the *l*th term of $X_i$.

$$\delta^{W}(x_{il}, z_{jl}) = \begin{cases} 0, & if \ x_{il} = z_{jl} \\ w_l^{\beta}, & if \ x_{il} \neq z_{jl} \end{cases} \tag{8}$$

In addition to improving the algorithm itself, some optimization algorithm is also used to optimize the FKM algorithm. For example, the genetic fuzzy k-modes algorithm integrated the genetic algorithm and the conventional fuzzy k-modes algorithm, which is called GFKM [36]. This algorithm has five basic steps: (1) string representation, (2) population initialization, (3) selection process, (4) crossover process, and (5) mutation process. The fuzzy k-modes algorithm which is optimized by the genetic algorithm can obtain the globally optimal solution and speed up the process of the convergence.

### 2.2. Consensus Clustering

Consistent clustering is a framework for clustering multiple algorithms or the same algorithm under different parameters to obtain better results. As shown in Figure 1, let $X = \{x_1, x_2, ..., x_n\}$ represents the dataset, and arbitrary cluster algorithms are used $p$ times to get $p$ different basic partitions $\pi_1, \pi_2, ..., \pi_p$ (BPs) [37].

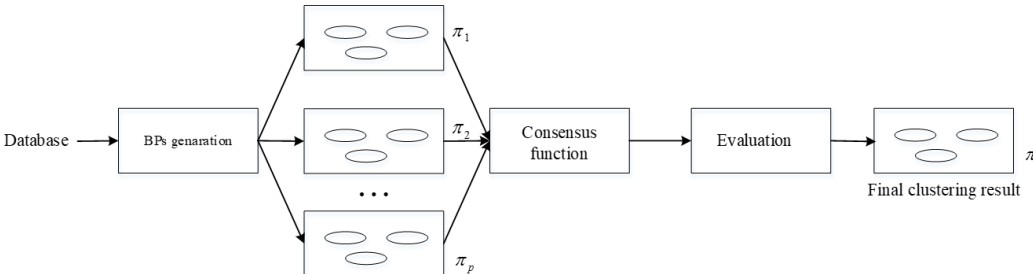

**Figure 1.** Consensus clustering framework

The number of clusters $K_1, K_2, ..., K_p$ in different partition is arbitrary. Each cluster result can transfer to the corresponding binary valued vector representation, and the binary value 1 indicates the sample $i$ belonging to the cluster, otherwise 0. Then, the consensus function needs to aggregate the BPs and obtains the final clustering result $\pi$. In this paper, the revised k-means algorithm which optimized by the DNA genetic algorithm is used as the consensus function. The DNA genetic algorithm is used to optimize the initial clustering center of the k-means algorithm, and the clustering quality can be evaluated by some common evaluation indicators, such as normalized mutual information, accuracy and F_measure, etc. The consensus clustering result is typically better than that obtained by the best BP. The following notations can be used to illustrate it:

$$\pi_i \equiv partition\ i : (k_i, C_1^i...C_{k_i}^i)$$

$$k_i \equiv number\ of\ clusters\ in\ partition\ \pi_i$$

$$C_j^i = \{s_l : s_l \in cluster\ j\ of\ partition\ \pi_i\}$$

$$\equiv list\ of\ samples\ in\ the\ jth\ cluster\ of\ partition\ \pi_i \tag{9}$$

$$X_j^i : X_j^i(k) = \begin{cases} 1 & if\ s_k \in C_j^i, k = 1,...,n \\ 0 & otherwise \end{cases}$$

$$\equiv binary\ valued\ vector\ representation\ of\ cluster\ C_j^i$$

*2.3. Chain and Hypergraph Structure*

A simple complex $S$ is a set of non-empty simplex $s_1, s_2, ..., s_p$. If $s_1 \prec s_2$, $s_1$ represent the vertex or face of the simplex $s_2$. Each complex also should be oriented. Therefore, a $S$-chain is a simplicial complex with $p$ dimensional simplices, which defined in [33]. All simplexes which have the same directions can combine into a chain domain.

A hypergraph $H = (v, e)$ represents a graph whose edge contains an arbitrary number of vertices [38–40]. $v$ is vertices sets and $e$ is hyper-edges sets. A hyper-edge can contain more than two vertices and can be formally represented by a nonempty subset of $v$. As shown in Figure.1. A hypergraph can better represent the complex information and the local group information than the traditional graph. In the traditional graph, there is only one edge between the two vertices if they are similar. However, in the hypergraph, we can construct a hyper-edge to connect more than two vertices. Therefore, the local group information and complex relationship hidden in the data can be captured by the hypergraph model [41].

We also can represent the hypergraph $H = (v, e)$ in an accessible matrix:

$$H = \begin{cases} 1, & v \in e \\ 0, & otherwise \end{cases}$$

where, $H = 1$ if the hyper-edge $e$ contains the vertex $v$. So, the Figure 2 can be expressed as:

$$\begin{bmatrix} & e_1 & e_2 & e_3 & e_4 \\ v_1 & 1 & 0 & 0 & 0 \\ v_2 & 1 & 1 & 0 & 0 \\ v_3 & 1 & 1 & 1 & 0 \\ v_4 & 0 & 0 & 0 & 1 \\ v_5 & 0 & 0 & 1 & 0 \\ v_6 & 0 & 0 & 1 & 0 \\ v_7 & 0 & 0 & 0 & 0 \end{bmatrix}$$

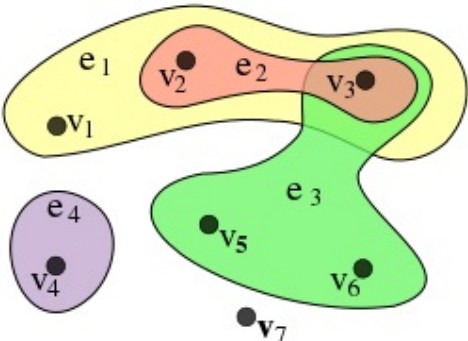

**Figure 2.** An example of the hypergraph which has vertices $v = \{v_1, v_2, v_3, v_4, v_5, v_6, v_7\}$, and hyper-edges $e = \{e_1, e_2, e_3, e_4\} = \{\{v_1, v_2, v_3\}, \{v_2, v_3\}, \{v_3, v_5, v_6\}\}$.

## 3. Coupling DNA-Chain-Hypergraph P System for Consensus Clustering (DCHP-FCC)

In this section, the coupling DCHP system is proposed. Firstly, the membrane structure of the DCHP system is introduced. Then, the different operations in the subsystem and consensus system are introduced, respectively. The flowchart of the proposed DCHP-FCC algorithm is shown in **Error! Reference source not found.**3.

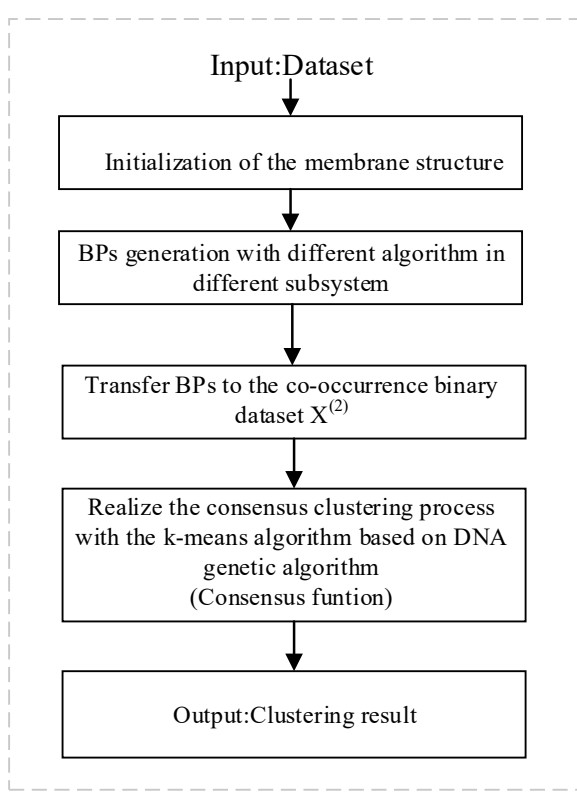

**Figure 3.** Flow chart of the proposed DNA-chain-hypergraph P System for consensus clustering (DCHP-FCC) algorithm.

### 3.1. Membrane Structure of the DCHP System

The DCHP system has two main membrane structures—the chain membrane and hyper-membrane structure. The basic framework of the chain membrane structure is shown in Figure 4 and the structure of the hyper-membrane structure is shown in Figure 5.

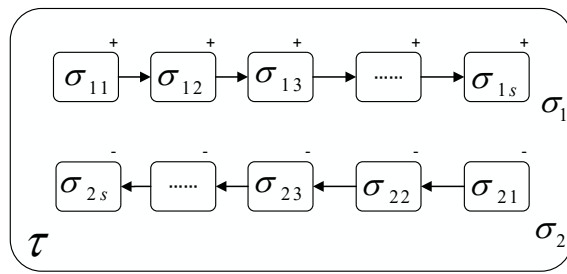

**Figure 4.** Chain membrane structure

**Definition 2.** *Based on the operation of the chain structure, the unit membrane can combine into chains. The chain membrane has two directions (i.e., "+" or "−"). As shown in Figure 4, the chain $\sigma_1$ is "+":* $\sigma_{11} \to \sigma_{12} \to \sigma_{13} \to \cdots \to \sigma_{1s}$ *and $s$ is the length of the chain $\sigma_1$. As in $\sigma_2$, the chain $\sigma_2$ is "−":* $\sigma_{21} \to \sigma_{22} \to \sigma_{23} \to \cdots \to \sigma_{2s}$. *Membranes $\sigma_{11}, \sigma_{12}, ..., \sigma_{1s}, \sigma_{21}, ..., \sigma_{2s}$ are all unit membranes. There is a channel between the adjacent unit membrane. $\tau$ is the skin membrane, and it is also the max membrane in the system. $\sigma_{11}, \sigma_{12}, ..., \sigma_{1s}, \sigma_{21}, ..., \sigma_{2s}$ are all called children membrane of the skin membrane $\tau$, so $\sigma_{11}, \sigma_{12}, ..., \sigma_{1s}, \sigma_{21}, ..., \sigma_{2s} \prec \tau$. A membrane without any other membranes inside it is an elementary membrane, so $\sigma_{11}, \sigma_{12}, ..., \sigma_{1s}, \sigma_{21}, ..., \sigma_{2s}$ are also represent elementary membrane.*

**Definition 3.** *Based on the topology structure of the hypergraph, the hyper-membrane structure is designed as a membrane with two or more upper membranes. In this system, it also has a similar definition to the chain membrane. For example, a membrane without any upper membrane is as skin membrane, and a membrane without any children membranes is an elementary membrane. For the two membranes $m_1$ and $m_2$, $m_1$ is the upper membrane of the $m_2$ if $m_2 \subset m_1$. If there is no membrane, $m_3$ satisfies $m_2 \subset m_3 \subset m_1$, and the membrane $m_2$ is the correspondingly lower membrane of $m_1$ As shown in Figure 5, membrane 1 is a skin membrane, and membranes 2, 3, 6, 8, 9, 10, 11 are all elementary membranes. In particular, membrane 8 is a hyper-membrane which has two upper membranes, 4 and 7.*

According to the basic membrane structure in Figure 4 and 5, a novel P system is designed for the consensus clustering process. The membrane structure of the DCHP system is shown in Figure 6. The DCHP system of degree $m > 0$ is defined as:

$$\prod = (O, \mu, \omega_1, \omega_2, ..., \omega_m, subsys_i, consys, i_0)$$

where:

- $O$ is the finite set of objects;
- $\mu$ represents the structure of the membrane. It includes the structure of the chain membrane, hyper-membrane and consensus membrane;
- $\omega_1, \omega_2, ..., \omega_m$ are objects in $O$, which represent the initial multisets objects in $m$ membranes at the beginning of the calculation; we denote the number of chain membrane is $m_1$, the number of hyper-membrane is $m_2$ and the number of membrane in consensus system is $m_3$. $m_1 + m_2 + m_3 = m$. $\lambda$ means the membrane has no object.
- $subsys_i$ is the subsystem which is used to generate the basic partition of clustering. In this system, three subsystems execute three kind of clustering algorithm, respectively.
- $consys$ is the consensus clustering membrane, which is used to generate the final clustering result.
- $i_0$ is the output membrane of the system $\prod$.

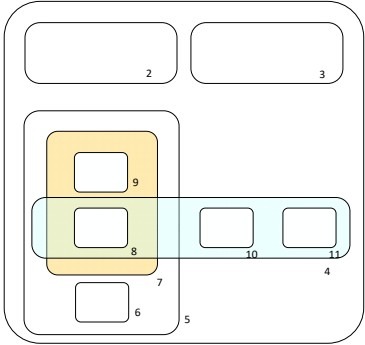

**Figure 5**. Hyper-membrane structure of the P system

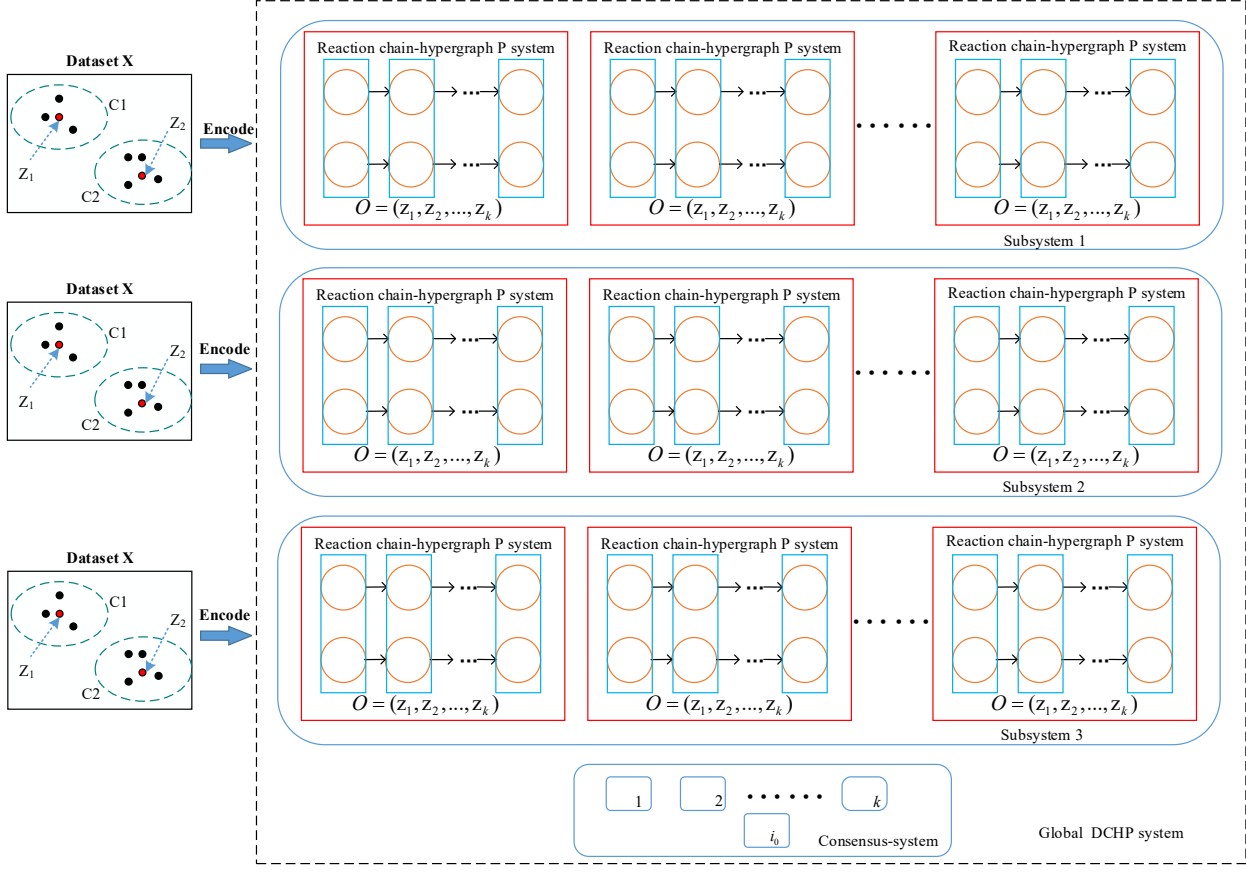

**Figure 6.** The membrane structure of the DCHP system

*3.2. The Consensus Clustering Realized with the DCHP System*

To implement the consensus clustering of the fuzzy k-modes algorithm, we propose three kinds of subsystem (i.e., reaction chain-hyper P system, consensus system, and global DCHP system). As shown in Figure 6, three classic algorithms (FKM [1], WFKM [2,16] and GFKM [36]) are simultaneously implemented in the three subsystems. The fuzzy k-modes algorithm generates a fuzzy partition matrix for the categorical data, and gives confidence to objects in different clusters. We call it soft partition, which is closer to reality than hard partition. This method is equal to all variables that determine cluster membership. However, the situation in the real world is usually different. The WFKM algorithm introduces a weight vector in the traditional FKM algorithm. This modification associates higher weights with the features which are instrumental in recognizing the clustering pattern of the data. These two methods are only improvements of the K-mode algorithm itself, and there are still some drawbacks in the convergence speed and the global optimization of the algorithm. In order to speed up the convergence process, the GFKM algorithm is proposed which used a one-step crossover process, mutation process and selection process in the clustering process.

### 3.2.1. Reaction Chain-Hypergraph P System in Subsystem

Initially, objects are randomly generated in the reaction chain hypergraph P system. The object represents a set of cluster centers. Suppose the dataset $X = \{X_1, X_2, ..., X_n\} \subseteq R^{n \times d}$ has $K$ clusters, where $n$ is the number of data points and $d$ is the dimension. The initial cluster centers $K_i (i = 1, 2, 3)$ are randomly selected out of $n$ data as the initial cluster centers and are denoted as $Z = \{z_1, z_2, ..., z_{k_i}\}$, where $z_i = \{z_{i1}, z_{i2}, ..., z_{id}\} \subseteq R^d$. $Z = \{z_1, z_2, ..., z_{k_i}\}$ is the initial string in the reaction chain hypergraph P system. Different reaction chain hypergraph P systems in the same subsystem conduct the same k-modes algorithm. Different subsystems conduct different K-modes algorithms with different parameters in parallel.

Then, each reaction chain-hyper P system generates a clustering result with different cluster number $K_i$. The clustering result $\pi_i$ is transferred to the corresponding subsystem. These results are also called basic partitions (BPs). Next, the object co-occurrence strategy is used for these BPs, and the details of this method can be seen in [42]. The BPs are transformed into a binary dataset $X^{(2)} = \{x_l^{(2)} \mid l = 1, 2, ..., n\} : x_l^{(2)} = < x_{l,1}^2, ..., x_{l,i}^2, ..., x_{l,p}^2 >, with\ x_{l,i}^{(2)} = < x_{l,i1}^2, ..., x_{l,ij}^2, ..., x_{l,iK_i}^2 >$, and

$$x_{l,ij}^{(2)} = \begin{cases} 1, if\ x_l\ belongs\ to\ cluster\ j\ in\ BP\ \pi_i \\ 0, otherwise \end{cases} \tag{10}$$

### 3.2.2. Local Communication Membrane System

Afterwards, the transformed binary dataset $X^{(2)}$ is transferred to the consensus system by the communication rule:

$$(p, a_k / q, \lambda) \tag{1}$$

where $a_k$ is the basic partition result in the subsystem and $\lambda$ means there is no object in the membrane. $p$ is the subsystem, and $q$ is the consensus system.

### 3.2.3. Consensus System

The revised k-means algorithm which is optimized by the DNA genetic algorithm is used as the consensus function. The DNA genetic algorithm is used to optimize the initial clustering center of the k-means algorithm. When the dataset $X^{(2)}$ appears in the consensus system, it is transferred to binary. So, we can use these values as the initial population. Then, the specific selection, crossover and mutation process of DNA genetic algorithm are described as follows:

(1) Selection operation

The optimal individuals with the first 10% fitness are directly inherited to the next generation, and the rest of the individuals are selected according to the roulette selection strategy.

(2) Crossover and mutation operation

One-point crossover and adaptive mutation operation are used in this step, respectively. The crossover probability is set as $P_c$. The mutation probability can be adjusted according to the fitness value of the individual. The specific mutation probability is updated as follows:

$$P_m = (f_{max} - f)/(f_{max} - f_{avg}) \qquad (2)$$

where, $P_m$ is the mutation probability, $f_{max}$ refers to the maximum fitness in every generation, $f_{avg}$ is the average fitness in every generation, and $f$ is the fitness of the individual. If the fitness is equal to the maximum fitness, the mutation probability is 0. This operation can guarantee that the optimal individual does not change by the mutation operation.

(3) Fitness function

The absolute-deviation criterion in the following is used to measure the clustering quality:

$$F = \sum_{k=1}^{K} \sum_{x_i \in C_k} dist(x_i, o_k) \qquad (3)$$

where, $x_i$ is data point $i$ in the dataset $X^{(2)}$, $o_k$ is the cluster center of the $C_k$. $dist(x_i, o_k)$ calculates the Euclidean distance between the data point $x_i$ and the corresponding cluster center $o_k$ .

The computation process can stop when the predefined maximum iteration is reached or the difference between two adjacent iterations is less than the given threshold $\varepsilon$. Then, the DCHP system is stopped, and the final results are output in the membrane $i_0$.

## 4. Experiments and Discussions

### 4.1. Data Sets and Parameter Settings

Seven datasets which were collected from the UCI Machine Learning Repository [43] are used in this section. The data type of all seven datasets is categorical data. These datasets are used in the comparison algorithms. Although the comparison algorithms used several other data sets in addition to these seven datasets, considering that these datasets have poor results in the comparison algorithms, they are not used as experimental data in this paper. Table 1 shows the detail information of these datasets. To generate the BPs, three different clustering algorithms are used. The number of clusters $K_i(i=1,2,3)$ in the process of BPs generation is set to $\left\lceil K_a, \sqrt{n} \right\rceil$, and $K_a$ is the actual number of clusters of the dataset. The cluster number of the consensus clustering process is set as $K_a$. All experiments are simulated by the MATLAB R2014b running on a Windows 7 platform of the 64-bit edition. The PC has an Intel Core i7-4770 3.4 GHZ CPU and 8 GB of RAM.

**Table 1.** Benchmark datasets.

| Dataset | # of Instance | # of Attributes | # of Classes |
|---|---|---|---|
| Soybean-small | 47 | 35 | 4 |
| Spect Heart | 267 | 22 | 2 |
| Tic-tac-toe | 958 | 9 | 2 |
| Voting | 435 | 16 | 2 |
| Breast cancer | 286 | 9 | 2 |
| Zoo | 101 | 17 | 7 |
| Mushroom | 8124 | 22 | 2 |

### 4.2. Evaluation Metric

To evaluation the performance of the DCHP-FCC algorithm, three external clustering evaluation metrics are selected [44], i.e., Adjusted Rank Index (ARI), Clustering Accuracy (ACC), and F_measure (F).

The ARI is defined as follows:

$$ARI(T,C) = \frac{2(ad - bc)}{(a+b)(b+d) + (a+c)(c+d)} \tag{4}$$

where, $T$ represents the pre-determined clustering label, and the $C$ represents the label of the clustering result. $a, b, c$ and $d$ refer to: (1) in the same class as $T$ and $C$, (2) in the same class as $T$ but not in the same class as $C$, (3) in the same class as $C$ but not in the same class as $T$, (4) in a different class $C$ and $T$, respectively.

The ACC can be calculated by:

$$ACC = \frac{\sum_{k=1}^{n} \delta(l_k, map(g_k))}{n} \tag{5}$$

where, $n$ is the number of data points, $g_k$ represents the clustering result labels which are obtained by the algorithm, and $l_k$ is the true class label of the data $x_i$. $map(\cdot)$ is the mapping function that maps the clustering labels obtained by the algorithm to the real clustering labels. When $l_k = map(g_k)$, the function value of $\delta(l_k, map(g_k))$ is equal to 1, and otherwise it is equal to 0.

The F_measure is defined as:

$$F\_measure(k,l) = \frac{2 \times (P(k,l) \times R(k,l))}{(P(k,l) + R(k,l))} \tag{6}$$

where, $P(k,l) = s_{kl}/s_k$ and $R(k,l) = s_{kl}/s_l$. $P(k,l)$ refers to the precision of cluster $k$ with respect to class $l$, and $R(k,l)$ represents the recall of cluster $k$ with respect to class $l$. $s_{kl}$ refers to data points which belong to both cluster $k$ and $l$, $s_k$ represents the number of data points in $k$, and $s_l$ is the number of data points in $l$.

### 4.3. Experiment Results and Analysis

In this section, we firstly select the number of BPs. According to the analysis in previous research [37], we can select the number of BPs as about 50, and the clustering effect gradually stabilizes. In this paper, we also need to determine the number of BPs, but we do not use the same BP generation strategy as in the other methods. In this paper, BPs need to be generated in each time. At the same time, considering the structure of the DCHP system and the characteristics of the three basic clustering algorithms, we need to guarantee that the BPs generated by the different algorithms are the same. So, the final number of BPs is a multiple of 3. Taking into account that the number of BPs is basically maintained at about 50, the clustering effect is the best, so we conduct preliminary experiments on the number of BPs with 30, 60 and 90, respectively. Experiments are run 30 times, and the mean and variance of the experimental results are recorded in each case. The experimental results in Table 2 show that when the number of BPs is 60, the experimental effect is the best.

**Table 2.** The preliminary experiments on the Soybean-small dataset.

| UCI Datasets | | DCHP-FCC(BPs = 90) | | | DCHP-FCC(BPs = 60) | | | DCHP-FCC(BPs = 30) | | |
|---|---|---|---|---|---|---|---|---|---|---|
| | | ARI | ACC | F | ARI | ACC | F | ARI | ACC | F |
| Soybean-small | Mean | 0.9338 | 0.9149 | 0.9052 | **0.9676** | **0.9567** | **0.9505** | 0.9228 | 0.8950 | 0.8853 |
| | Std. | 0.0069 | 0.0112 | 0.0140 | **0.0044** | **0.0078** | **0.0102** | 0.0086 | 0.0145 | 0.0180 |

At the same time, the boxplots for the 30 times are also shown in Figure 7. In Figure 7, (a) is the result of the NMI metric, (b) is the result of the ACC metric, and (c) is the result of the F_measure metric. The red line is the center of the box. The top line is the third quartile and the bottom line is the first quartile of the box. The upper and lower limits of the whiskers represent the maximum and the minimum values, respectively. The symbol '+' represents the outlier. Note that the better clustering result, the higher and more compact the boxes are. We can see from Figure 7, when the BPs is 60, the best clustering results are obtained. Therefore, the number of BPs in subsequent experiments is determined to be 60.

Next, the DCHP-FCC algorithm is compared with three basic clustering algorithms and one improved algorithm, IWFKM, which was proposed in [16]. In order to maintain the originality of the benchmark algorithms, this paper obtains the results of these algorithms through their own algorithms and parameters, which are shown in Table 3. The mutation probability in the GFKM algorithm is also set as 0.01. For comparison, the maximum number of iterations of the comparison algorithm is 100.

**Table 3.** Parameter setting of the proposed algorithm and the comparison algorithms.

| Algorithm | Parameters | Description | Setting |
|---|---|---|---|
| **FKM** | $K_1$ | The number of clusters | Random select in $\left[ K_a, \sqrt{n} \right]$ |
| **WFKM/IWFKM** | $K_2$ | The number of clusters | Random select in $\left[ K_a, \sqrt{n} \right]$ |
| **GFKM** | $K_3$ | The number of clusters | Random select in $\left[ K_a, \sqrt{n} \right]$ |
| | Pop_size | Population size | 100 |
| | Max_iter | Maximum number of generations | 100 |
| | $P_m$ | Mutation parameter | 0.01 |
| **Consensus clustering** | $K_a$ | The number of clusters | Real number of clusters |
| | $\varepsilon$ | Iteration stopping criteria | $e^{-5}$ |
| | $P_c$ | Crossover parameter | 0.7 |
| | $P_{m1}$ | Mutation parameter | 0.01 |

Every experiment is also run 30 times, and the mean and variance of the experimental results are recorded in each case. The experimental results are shown in Table 4. As we can see from Table 4, the performance of the DCHP-FCC is much better than the compared algorithms on Soybean-small, Spect heart, Voting, Zoo and Mushroom datasets, and partly better than the compared algorithms on the Tic-tac-toe and Breast cancer datasets. For the Tic-tac-toe and Breast cancer datasets, the DCHP-FCC algorithm can obtain the best ARI value, and the GFKM algorithm attains the best ACC and F_measure values. Even though the DCHP-FCC algorithm does not perform best on the Tic-tac-toe and Breast cancer datasets, the metric value is only 0.01 lower than the best value. So, the DCHP-FCC algorithm is more effective in dealing with the categorical clustering than the compared clustering algorithms.

**Table 4.** The comparison of ARI, ACC and F-measure (F) benchmark for four clustering algorithms on seven datasets (mean ± std., BPs = 60).

| UCI Datasets | | DCHP-FCC | FKM | WFKM | GFKM | IWFKM |
|---|---|---|---|---|---|---|
| Soybean-small | ARI | **0.97 ± 0.004** | 0.86 ± 0.006 | 0.88 ± 0.010 | 0.89 ± 0.009 | 0.87 ± 0.006 |
| | ACC | **0.96 ± 0.008** | 0.83 ± 0.009 | 0.86 ± 0.015 | 0.86 ± 0.013 | 0.85 ± 0.010 |
| | F | **0.95 ± 0.010** | 0.80 ± 0.012 | 0.84 ± 0.021 | 0.84 ± 0.017 | 0.83 ± 0.015 |
| Spect Heart | ARI | **0.75 ± 0.002** | $0.50 ± 4 \times 10^{-5}$ | $0.50 ± 2 \times 10^{-5}$ | 0.51 ± 0.001 | $0.50 ± 4 \times 10^{-5}$ |
| | ACC | **0.79 ± 5 × $10^{-32}$** | 0.79 ± 5 × $10^{-32}$ | 0.79 ± 5 × $10^{-32}$ | 0.79 ± 5 × $10^{-32}$ | 0.79 ± 5 × $10^{-32}$ |
| | F | **0.74 ± 0.002** | 0.63 ± 0.000 | 0.63 ± 0.000 | 0.63 ± 0.003 | 0.63 ± 0.000 |
| Tic-tac-toe | ARI | **0.52 ± 0.000** | 0.51 ± 0.000 | $0.50 ± 3 \times 10^{-5}$ | **0.52 ± 0.001** | $0.51 ± 5 \times 10^{-5}$ |

| | | | | | | |
|---|---|---|---|---|---|---|
| | ACC | $0.65 \pm 2 \times 10^{-31}$ | $0.65 \pm 4 \times 10^{-5}$ | $0.65 \pm 2 \times 10^{-31}$ | $\mathbf{0.66 \pm 0.000}$ | $0.65 \pm 2 \times 10^{-31}$ |
| | F | $0.59 \pm 0.001$ | $0.57 \pm 0.001$ | $0.55 \pm 0.001$ | $\mathbf{0.60 \pm 0.002}$ | $0.56 \pm 0.001$ |
| **Voting** | ARI | $\mathbf{0.78 \pm 0.000}$ | $0.75 \pm 0.000$ | $0.75 \pm 3 \times 10^{-6}$ | $0.75 \pm 0.000$ | $0.75 \pm 2 \times 10^{-6}$ |
| | ACC | $\mathbf{0.88 \pm 0.000}$ | $0.86 \pm 0.000$ | $0.85 \pm 1 \times 10^{-6}$ | $0.86 \pm 0.000$ | $0.85 \pm 1 \times 10^{-6}$ |
| | F | $\mathbf{0.88 \pm 0.000}$ | $0.86 \pm 0.000$ | $0.85 \pm 1 \times 10^{-6}$ | $0.86 \pm 0.000$ | $0.85 \pm 1 \times 10^{-6}$ |
| **Breast cancer** | ARI | $\mathbf{0.51 \pm 1 \times 10^{-4}}$ | $0.50 \pm 6 \times 10^{-6}$ | $0.50 \pm 3 \times 10^{-5}$ | $0.51 \pm 7 \times 10^{-4}$ | $0.50 \pm 6 \times 10^{-6}$ |
| | ACC | $0.70 \pm 5 \times 10^{-32}$ | $0.70 \pm 5 \times 10^{-32}$ | $0.70 \pm 5 \times 10^{-32}$ | $\mathbf{0.71 \pm 5 \times 10^{-5}}$ | $0.70 \pm 5 \times 10^{-32}$ |
| | F | $0.58 \pm 8 \times 10^{-4}$ | $0.56 \pm 3 \times 10^{-4}$ | $0.55 \pm 7 \times 10^{-4}$ | $\mathbf{0.59 \pm 0.002}$ | $0.54 \pm 4 \times 10^{-4}$ |
| **Zoo** | ARI | $\mathbf{0.90 \pm 0.000}$ | $0.87 \pm 0.001$ | $0.87 \pm 0.002$ | $0.89 \pm 0.002$ | $0.86 \pm 0.001$ |
| | ACC | $\mathbf{0.87 \pm 0.000}$ | $0.84 \pm 0.002$ | $0.83 \pm 0.003$ | $0.84 \pm 0.002$ | $0.83 \pm 0.003$ |
| | F | $\mathbf{0.78 \pm 0.004}$ | $0.74 \pm 0.005$ | $0.73 \pm 0.008$ | $0.77 \pm 0.004$ | $0.72 \pm 0.006$ |
| **Mushroom** | ARI | $\mathbf{0.81 \pm 1 \times 10^{-31}}$ | $0.67 \pm 0.014$ | $0.70 \pm 0.014$ | $0.68 \pm 0.016$ | $0.67 \pm 0.018$ |
| | ACC | $\mathbf{0.89 \pm 2 \times 10^{-31}}$ | $0.76 \pm 0.017$ | $0.80 \pm 0.014$ | $0.74 \pm 0.015$ | $0.76 \pm 0.017$ |
| | F | $\mathbf{0.89 \pm 2 \times 10^{-31}}$ | $0.76 \pm 0.015$ | $0.79 \pm 0.014$ | $0.77 \pm 0.016$ | $0.76 \pm 0.019$ |

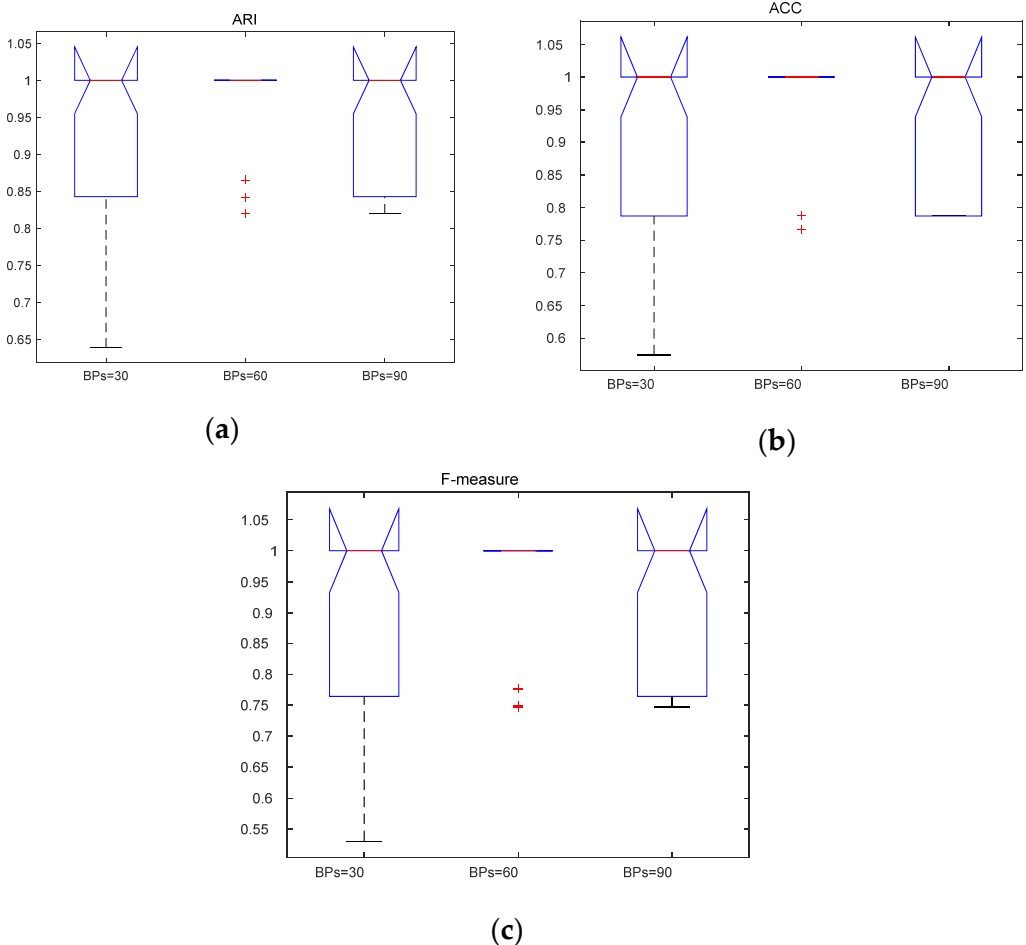

**Figure 7.** Boxplots of Adjusted Rank Index (ARI), Clustering Accuracy (ACC), and F_measure by DCHP-FCC algorithm on Soybean-small dataset with basic partitions (BPs) equal to 30, 60 and 90, respectively.

*4.4. Significance Testing*

In this section, the hypothetical tests on the values of seven UCI datasets are computed between the DCHP-FCC algorithm and other four comparison clustering algorithms. The results are shown in Tables 5–7. The significance level is set as $\rho < 0.05$. According to the results in Tables 5–7, the symbol '+' represents when the difference between the DCHP-FCC algorithm and the compared algorithm is significant, and the symbol '−' represents when the difference between the DCHP-FCC algorithm and the compared algorithm is not significant. We can see from Tables 5–7 that the results of the hypothetical test are almost '+'. This proves that there is a clear difference between the algorithm in this paper and the comparison algorithms.

**Table 5.** The *p*-value produced by the *T*-test in terms of ARI.

| UCI Datasets | DCHP-FCC vs. | | | |
|---|---|---|---|---|
| | FKM | WFKM | GFKM | IWFKM |
| **Soybean-small** | $0.9 \times 10^{-8}$(+) | $5.0 \times 10^{-5}$(+) | $3.0 \times 10^{-4}$(+) | $6.6 \times 10^{-6}$(+) |
| **Spect Heart** | $5.7 \times 10^{-27}$(+) | $1.0 \times 10^{-23}$(+) | $5.1 \times 10^{-22}$(+) | $1.1 \times 10^{-23}$(+) |
| **Tic-tac-toe** | $7.0 \times 10^{-14}$(+) | $1.8 \times 10^{-14}$(+) | $3.5 \times 10^{-10}$(+) | $8.5 \times 10^{-5}$(+) |
| **Voting** | $6.1 \times 10^{-9}$(+) | $1.8 \times 10^{-12}$(+) | $1.5 \times 10^{-6}$(+) | $6.9 \times 10^{-13}$(+) |
| **Breast cancer** | $6.6 \times 10^{-4}$(+) | $0.0077$(+) | $0.3373$(−) | $3.1 \times 10^{-4}$(+) |
| **Zoo** | $0.0092$(+) | $0.0033$(+) | $0.3104$(−) | $0.0020$(+) |
| **Mushroom** | $3.4 \times 10^{-7}$(+) | $2.6 \times 10^{-5}$(+) | $5.1 \times 10^{-6}$(+) | $4.9 \times 10^{-6}$(+) |

**Table 6.** The *p*-value produced by the *T*-test in terms of ACC.

| UCI Datasets | DCHP-FCC vs. | | | |
|---|---|---|---|---|
| | FKM | WFKM | GFKM | IWFKM |
| **Soybean-small** | $2.9 \times 10^{-7}$(+) | $1.1 \times 10^{-4}$(+) | $8.4 \times 10^{-4}$(+) | $2.2 \times 10^{-5}$(+) |
| **Spect Heart** | - | - | - | - |
| **Tic-tac-toe** | - | - | - | - |
| **Voting** | $7.2 \times 10^{-9}$(+) | $1.2 \times 10^{-12}$(+) | $1.8 \times 10^{-6}$(+) | $4.3 \times 10^{-13}$(+) |
| **Breast cancer** | - | - | - | - |
| **Zoo** | $5.5 \times 10^{-4}$(+) | $0.0027$(+) | $0.0060$(+) | $4.8 \times 10^{-5}$(+) |
| **Mushroom** | $0.0414$(+) | $0.5516$(−) | $0.1215$(−) | $7.9 \times 10^{-6}$(+) |

**Table 7.** The *p*-value produced by the *T*-test in terms of F_measure.

| UCI Datasets | DCHP-FCC vs. | | | |
|---|---|---|---|---|
| | FKM | WFKM | GFKM | IWFKM |
| **Soybean-small** | $4.3 \times 10^{-7}$(+) | $2.0 \times 10^{-4}$(+) | $8.7 \times 10^{-4}$(+) | $1.1 \times 10^{-4}$(+) |
| **Spect Heart** | $1.9 \times 10^{-14}$(+) | $3.7 \times 10^{-14}$(+) | $1.2 \times 10^{-9}$(+) | $8.1 \times 10^{-13}$(+) |
| **Tic-tac-toe** | $0.0540$(−) | $6.0 \times 10^{-6}$(+) | $0.5474$(−) | $1.1 \times 10^{-4}$(+) |
| **Voting** | $6.8 \times 10^{-9}$(+) | $1.1 \times 10^{-12}$(+) | $1.6 \times 10^{-6}$(+) | $3.9 \times 10^{-13}$(+) |
| **Breast cancer** | $9.7 \times 10^{-4}$(+) | $1.9 \times 10^{-4}$(+) | $0.3499$(−) | $1.7 \times 10^{-6}$(+) |
| **Zoo** | $0.0374$(+) | $0.0280$(+) | $0.3151$(−) | $0.0023$(+) |
| **Mushroom** | $2.3 \times 10^{-6}$(+) | $8.8 \times 10^{-5}$(+) | $1.0 \times 10^{-5}$(+) | $6.6 \times 10^{-6}$(+) |

**5. Conclusions**

In this paper, we propose a novel P system (DCHP) with a hybrid structure which combines the advantage of the chain structure and hypergraph topology structure for the consensus fuzzy k-modes clustering. The DCHP system has three subsystems and one consensus system. The subsystems are used to generate three kinds of basic partitions, respectively, and the consensus system is used to

realize the consensus clustering process with evolution operations. Then, the DCHP-FCC algorithm is compared with four k-modes clustering algorithms on seven UCI datasets and uses three performance validation indices: ARI, ACC and F_measure. The experimental results show that the DCHP-FCC algorithm can get better clustering results than the other compared algorithms.

There are several ways to continue this study in the future. Firstly, we can consider using different basic clustering algorithms in the consensus clustering or design a new P system structure instead of DCHP system. Secondly, the algorithm which can identify the optimal number of clusters should be investigated for categorical data in the consensus clustering process. In addition, the other clustering method can be used in the consensus clustering process.

**Author Contributions:** Conceptualization, Z.J. and X.L.; Methodology, Z.J. and X.L.; Software, Z.J.; Validation, Z.J.; Formal Analysis, Z.J.; Writing—Original Draft Preparation, Z.J.; Writing—Review & Editing, Z.J. and X.L.; Supervision, X.L.; Project Administration, X.L.; Funding Acquisition, X.L. All authors have read and agreed to the published version of the manuscript.

**Funding:** This work was partially supported by the National Natural Science Foundation of China (Nos. 61876101, 61802234 and 61806114), the Social Science Fund Project of Shandong (16BGLJ06, 11CGLJ22), China Postdoctoral Science Foundation Funded Project (2017M612339, 2018M642695), Natural Science Foundation of the Shandong Provincial (ZR2019QF007), China Postdoctoral Special Funding Project (2019T120607) and Youth Fund for Humanities and Social Sciences, Ministry of Education (19YJCZH244).

**Conflicts of Interest:** The authors declare no conflict of interest.

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
