# Peer review of "A Novel Consensus Fuzzy K-Modes Clustering Using Coupling DNA-Chain-Hypergraph P System for Categorical Data"

_processes, doi:10.3390/pr8101326_

Round 1
Reviewer 1 Report
When comparing the results obtained for all methods, were optimal clustering parameters found for each of them?
Has the use of other clustering algorithms such as k-harmonic means been considered?
The standard deviation of some of the results in Table 3 is 0, which would indicate deterministic results. However, there is probably a variation in the results.
Author Response
Thanks for your suggestion. We have revised the manuscript according to your suggestion. The specific reply is display in PDF.

Reviewer 2 Report
This article requires an extensive review in English.
Often, the article presents important grammatical errors and makes reading difficult.
Some of the sentences, like the following, require additional information: "The weight fuzzy k-mode algorithm introduces the weight vector".
The consensus clustering framework lacks important details. For example, it does not explain the consensus function.
The purpose of Formula 10 is unclear.
Experiments supported by Evolutionary Algorithms require additional details, in terms of configuration and results.
Moreover, the authors did not provide any justification for the selection of the selected data sets and their impact on categorical data.
Formula 18 lacks the introduction of "measure" and the final parenthesis.
The following sentence should refer to "l" instead of "k" "s_l is the number of data points in k". "S" must also be displayed in the denominator.
Author Response

(The authors gave the same response as above.)

Reviewer 3 Report
Authors proposed a method for categorical data. Although the scope of the paper is interesting, there are some major problems.
- One of the main problems of this research is its “literature review”. This section is very week. There are numerous methods for categorical data clustering and each year many papers are published. Authors should critically review these articles.
- In addition, authors should clearly define why your method is novel.
- There are many sentences that are not professional. in addition, the paper's writing should be enhanced. Further proofreading by a professional English editor is needed.
- Another problem of this paper is methods that have been used for testing the proposed approach. As I mentioned before, there are many methods in the literature, I highly recommend the authors to compare their method with them.
- Please revise the abstract and highlight the contribution of your research in the abstract.
- Please clearly highlight research gaps and your contributions.
- The steps of the proposed method are not well written and is a bit hard for the reader to follow. Please revise this section.
- As far as I know, you can find more data set, why did you select these data set?
Author Response

(The authors gave the same response as above.)

Round 2
Reviewer 2 Report
Reviewer #2 Concern #1
English requires an extensive editing revision, such as in the following sentence:
"Thereafter, there are many algorithm are proposed for the categorical data clustering".
or frequently on conjugating verbs, like the one in this case:
"We design three subsystem and one consensus system to implement consensus clustering.".
Reviewer #2 Concern #3
Still not clear the relevance of this formula, given that it has no applicability in the rest of this paper.
Reviewer #2 Concern #5:
The introduced justification was not included in the paper. As an alternative, a reference to similar works supported by the same datasets can help.
Author Response
Reviewer #2 Concern #1
English requires an extensive editing revision, such as in the following sentence:
"Thereafter, there are many algorithm are proposed for the categorical data clustering".
or frequently on conjugating verbs, like the one in this case:
"We design three subsystem and one consensus system to implement consensus clustering.".
Response: Thanks for your suggestions. We revised the full text of the paper again. At the same time, the two sentences pointed out by the reviewer are revised as follows
(1) Thereafter, many algorithms have been proposed for the clustering of categorical data.
(2) The DCHP-FCC system contains three reaction chain membrane subsystems and one consensus subsystem.
Reviewer #2 Concern #3
Still not clear the relevance of this formula, given that it has no applicability in the rest of this paper.
Response: Thanks for your suggestion. This formula is the explanation of the chain structure, but really do not used in the rest of the paper anymore. So we delete it. The revised content is shown as follows:
A simple complex is a set of non-empty simplex . If , represent the vertex or face of the simplex . Each complex also should be oriented. Therefore, a -chain is a simplicial complex with dimensional simplices, which defined in [24]. All simplexes which have same directions can combine into a chain domain.
Reviewer #2 Concern #5:
The introduced justification was not included in the paper. As an alternative, a reference to similar works supported by the same datasets can help.
Response: Thanks for your suggestion. We add the reason why we choose these kind of datasets in the experiment in the paper. The specific content is shown as follows :
The data types of seven datasets are all categorical data. These datasets are used in the comparison algorithms. Although the comparison algorithms used several other data sets in addition to these seven data sets, considering that these data sets have poor results in the comparison algorithms, so they are not used as experimental data in this paper.

Reviewer 3 Report
Thanks for your revision. However, still, I think some comments have not been adequately addressed. particularly following comments, please work on them:
- One of the main problems of this research is its “literature review”. This section is very week. There are numerous methods for categorical data clustering and each year many papers are published. Authors should critically review these articles.
- In addition, authors should clearly define why your method is novel.
- Another problem of this paper is methods that have been used for testing the proposed approach. As I mentioned before, there are many methods in the literature, I highly recommend the authors to compare their method with them.
Author Response
The modified contents are in the PDF
Round 3
Reviewer 3 Report
The current version is acceptable.